# Multicore Fiber Bending Sensors with High Sensitivity Based on Asymmetric Excitation Scheme

**DOI:** 10.3390/s22155698

**Published:** 2022-07-29

**Authors:** Lina Suo, Ya-Pei Peng, Nan-Kuang Chen

**Affiliations:** 1School of Physics Sciences and Information Technology, Liaocheng University, Liaocheng 252000, China; 1920110506@stu.lcu.edu.cn; 2College of Engineering Physics, Shenzhen Technology University, Shenzhen 518000, China; 3NK Photonics Ltd., Jinan 250119, China

**Keywords:** multicore fiber, four core fiber, supermodes, bending sensors, asymmetric modes

## Abstract

Bending sensing was realized by constructing a tapered four-core optical fiber (TFCF) sensor. The four-core fiber (FCF) between the fan-in and fan-out couplers was tapered and the diameter became smaller, so that the distance between the four cores arranged in a square became gradually smaller to produce supermodes. The two ends of the TFCF were respectively connected to the fan-in and fan-out couplers so that the individual cores in the FCF could link to the separate single-mode fibers. A broadband light source (superluminescent diodes (SLD)) spanning 1250–1650 nm was injected into any one of the four cores, and the orientation was thus determined. In the tapering process, the remaining three cores gradually approached the excitation core in space to excite several supermodes based on the tri-core structure first, and then transited to the quadruple-core structure. The field distributions of the excited supermodes were asymmetric due to the corner-core excitation scheme, and the interference thus resulted in a higher measurement sensitivity. When the diameter of the TFCF was 7.5 μm and the tapered length was 2.21 mm, the sensitivity of the bending sensor could reach 16.12 nm/m^−1^.

## 1. Introduction

With the continuous development of sciences and technologies, the demands for fiber sensors have increased drastically due to their merits of high signal capacity, high transmission bandwidth, high measurement accuracy, high detection sensitivity, and high environmental stability [1,2,3,4,5,6]. In practical applications, bending is one of the important physical parameters in mechanical engineering, material engineering, and structural health monitoring [7,8]. At present, fiber bending sensors are generally divided into the grating type and the interferometer type. It is known that the fiber Bragg gratings (FBGs) have been extensively employed in bending sensing due to their high accuracy and high sensitivity when the narrowband reflection spectrum changes with the varied grating period [9,10]. In addition, multiple FBGs can also be deployed simultaneously for wavelength-division multiplexing (WDM) or time-division multiplexing (TDM) configurations to improve the signal capacity for sensing. The lights are confined within the core for uniform FBGs, allowing them to be used for physical measurements of temperature, strain, or pressure. However, for special applications such as biosensing, interactions with the surrounding medium are required, and are usually obtained using cladding mode excitation [11,12]. On the other hand, fiber interferometers are also good options for bending sensing, since the angular displacement can lead to variations in the optical path difference (OPD) between the interfering optical beams to change the resonant wavelengths [13]. Except for the sensitivity and accuracy, the detection of multiple working parameters are also interesting topics. For instance, the combination of Bragg gratings and multicore fibers (MCFs) were proposed to achieve a sensitivity down to 10^−3^ m^−1^ [14,15]. However, the distance between the cores of MCFs are far enough to avoid of any evanescent coupling, and the MCFs mainly serve in the role of supporting the cores only. The concept of creating MCFs was proposed in the 1980s [16]. However, in contrast to those MCFs used for supporting hosts, strongly coupling MCFs are very different, and can be used to excite the supermodes to achieve high-sensitivity interferometric sensors. In fact, in order to meet the huge demands for transmission bandwidth in communication and sensing applications, various types of active and passive MCFs have been extensively studied in recent years to expand the signal capacity based on space-division multiplexing technology [17,18,19,20,21]. MCFs can also provide multiparameter sensing [22,23,24] or high-power lasers based on coherent beam combination [25]. MCF has different optical characteristics from ordinary optical fiber. For example, the evanescent wave optical trapping force and split evanescent fields have been found in seven-core optical fibers [26,27]. In addition, attenuated power coupling may occur when the MCF is severely bent [28]. Usually, the fan-in/fan-out couplers are essential to help transmit the optical signal into the individual cores of the MCF when the sensor is composed of MCF. When the MCF is tapered, supermodes [29,30,31,32] are excited, leading to higher sensitivity due to interference. Until now, sensors based on four-core fiber (FCF) have been reported for temperature, refractive index, and pressure measurements [33,34,35]. However, the reported sensors based on FCF were untapered, and the guided modes were excited through the silica cladding when the standard SMF was fusion-spliced to the center silica, within the areas surrounded by the four Ge cores [36]. This scheme of excitation method results in an abundance of high-order cladding modes.

In contrast, in this work, it was experimentally revealed that the asymmetric supermodes were more sensitive to the variations in bending based on the corner-core excitation method in the TFCF. Several supermodes can be excited when the locations of the multicores are close enough in tapering to make the evanescent fields overlap each other [29,30,31,32,37]. However, the optical power is not equally and concentrically distributed due to the corner-core excitation scheme. The mode fields of the excited supermodes are consequently asymmetric; microphotographs of the excited supermodes are shown in the next section to verify the phenomenon. Clearly, TFCF is superior to the conventional tapered fiber coupler when serving as fiber bending sensors. In this work, the excitation of the asymmetric supermodes in the TFCF was proposed to construct bending sensors. During tapering, the remaining three cores gradually approached the excitation core in space to excite several asymmetric supermodes based on the tri-core structure first, and then transited to the quadruple-core structure. The best bending sensitivity was up to 16.12 nm/m^−1^ within the bending range of 0.399–1.129 m^−1^, and the best extinction ratio was 16.85 dB when the diameter of the TFCF was 7 μm and the tapered length was 2.21 mm.

## 2. Experimental Setup and Working Principle

In this study, the bending sensor was fabricated by using TFCF, and the input signal was launched into core 1 of the FCF. Since the mode field distribution was asymmetric, the orientation of excitation was thus important to determine the direction with wider evanescent field penetration for achieving a high bending sensitivity. When the input port of the launching light was changed, the orientation of excitation, as well as the bending sensitivity, were changed accordingly. Subsequently, the asymmetric supermode interference could be generated when the FCF was tapered. During tapering, the evanescent field of the excitation core in the FCF extended to the evanescent fields region of the adjacent two cores to excite the new modes through evanescent coupling. All the new excited modes, together with the original guiding mode, are commonly supposed to form the asymmetric supermodes based on the three cores [38,39,40,41,42]. This phenomenon changes rapidly when the diagonal core enters the evanescent coupling region of the first three grouped cores to produce new asymmetric supermodes. Hence, the supermodes are generated from a tri-core structure to transit to a quadruple-core structure in a smaller TFCF. Figure 1a shows cross-sectional picture of the FCF under a 1000× CCD microscope. The four cores of the FCF (Chiral Photonics: SM-4C1500) were physically arranged in a square. The cladding diameter was 125 μm and the core diameter was 8 μm. The distance between adjacent cores was 50 μm, the distance between the two diagonal cores was 74 μm, and the numerical aperture of each core was 0.157. All the experiments were carried out in a clean room with constant temperature and humidity.

Firstly, a section of FCF with the coating layer removed was fixed on the optical fiber-tapering platform, then heated using a hydrogen flame at a heating temperature of approximately 900 °C. The FCF was then heated and stretched using a scanning flame to form a desired uniformed tapered diameter and tapered length. The precise displacement resolution of 1 μm of the stepping motors could be realized during tapering to reduce the diameter (D) of the FCF. Finally, the TFCF sensor was obtained. Figure 1b shows the experimental setup for constructing the TFCF and bending sensors. A broadband light source spanning 1250–1650 nm wavelengths from the superluminescent diodes (SLD) was launched into one of the cores in the FCF through a fan-in coupler. The mode excitation method through one of the four cores located in a rectangular configuration to produce asymmetric supermodes is referred to as a corner-core excitation scheme in this work. The spectral responses of the four cores were individually connected to the optical spectrum analyzer (OSA: YOKOGAWA AQ6370D, Tokyo, Japan) through the fan-out coupler to record the spectral characteristics. The benefit of using the TFCF to construct the interferometric bending sensors was that the asymmetric supermodes could be excited when the corner-core excitation method is employed. The asymmetric supermodes had evenly distributed mode fields, which made the evanescent field penetration extend more toward a certain direction to improve the bending sensitivity. In addition, the FCF is more popular and commercially available compared to other types of MCFs. The inset picture in Figure 1b shows the TFCF with a D of 37.5 μm. During tapering, the distance between the cores decreased, while the Gaussian mode expanded with the decreasing core diameter. These were the main reasons why the mode field distribution gradually expanded and the fields overlapped each other. In addition, the four Ge-doped cores were gradually blurred under a 1000× CCD microscope due to the diffusion of Ge in the core [43]. This gave rise to a reduced numerical aperture, as well as its *V*-value (*V*
$=\frac{2\pi a}{\lambda }\sqrt{n_{1}^{2}-n_{2}^{2}}$), where a, *n*_1_, *n*_2_, and *λ* are the core radius, refractive index of the core, refractive index of the cladding, and the wavelength, respectively, which led to a wider mode field diameter [44]. The SLD white light was launched into core 1, and the OSA was used to separately monitor and record the spectral response of the output ends of cores 1–4. When D was above 30 μm, the optical signal passed directly; that is, input from core 1 and output from core 1. When the supermodes based on the tri-core structure were excited and disturbed, the spectral response began to change, which was reflected in the extinction ratio (ER) of the oscillating output spectra. The ER of the oscillating output spectra increased with the decrease in D. In addition, with the decrease in D, the excitation based on the tri-core structure also rapidly transformed to the four-core structure. Figure 1c,d show the optical power distributions for the excited asymmetric supermodes when D = 36 μm and 23 μm, respectively. The mode patterns were clearly both asymmetric. In Figure 1c, the mode patterns of the supermodes were clearly excited based on cores 1–3. In contrast, in Figure 1d, the supermodes were produced by cores 1–4, since the D was smaller than that shown in Figure 1c. Obviously, the asymmetric mode field distributions of the supermodes were helpful to achieve a high bending sensitivity. For measurements, the tapered regions of the FCF were mounted and fixed on a 12 mm long thin plastic plate as the supporting material during curvature bending to avoid any extra linear tension/compression. Compared with cores 2–4, the excitation core 1 was located at the bottom, and the plastic plate was convex and bent upward. In future works, the high accuracy rotational stages can be further embedded at the mechanical clampers to investigate the orientation-dependent effect on curvature bending.

## 3. Theoretical Model

In order to confirm that the TFCF could excite the asymmetric supermode, some simulations were undertaken. The electric field distributions were simulated using Rsoft 2020 (version number: 2020.09, Synopsys, Mountain View, CA, USA) based on the beam propagation method, which considered a refractive index of the core and cladding of 1.4525 and 1.444, respectively, and a working wavelength of 1550 nm. The model for the FCF is shown in Figure 2a. It can be seen that the core and cladding parts of the outline were clear, with no overlapping phenomenon. According to the scale analysis shown in Figure 2a, the FCF core and cladding diameter, spacing, and other parameters were correctly set in this model. The light source excited the core in the upper right corner, which we called the corner-core excitation scheme, and the laser was well confined within the core, as shown in Figure 2b. The electric field distributions are shown in Figure 2c–h. When the TFCF gradually became thinner and the four fiber cores gathered more closely, the evanescent field in the excitation core was increasingly coupled with the two adjacent fiber cores. Then, the supermode was excited to form the structure with three fiber cores coupled, as shown in Figure 2c,d. As the diameter of the fiber continued to thin, the fourth core gradually entered the overlapping region of the asymptotic field relative to the diagonal position of the excitation core to stimulate the new supermode, thus forming a coupled four-core structure, as shown in Figure 2e–g. When the excitation light was coupled to the thinnest uniform region of the TFCF, there were many chaotic coupled modes in this uniform region because the original excitation was an asymmetric mode, as shown in Figure 2g. Then, as the light passed through the uniform region, there was a process of transmission back to the four cores as the fiber diameter gradually increased, as shown in Figure 2h. In summary, it was found through simulation that the corner-core excitation scheme could stimulate the asymmetric mode, the coupling process of which could be divided into the three-core coupling mode between the excitation core and two adjacent fiber cores and the four-core coupling mode formed by the oblique fiber core entering the evanescent field through three-core coupling.

## 4. Results and Discussion

During the measurement, the TFCF samples with a D of 26 μm, 13.5 μm, and 7.5 μm were prepared and identified as TFCF-1, TFCF-2, and TFCF-3, respectively, and their optical properties were studied. The corresponding uniformed tapered length L was 1 mm, 2.18 mm, and 2.21 mm, respectively. The L was observed and measured under a 1000× CCD microscope. The L was defined as the length of the region where the variation of D was less than 0.1 μm. The total tapered region contained the taper transition and was of course longer than L. In this work, the tapered fibers were respectively glued and mounted on a thin plastic plate with a 12 mm long region for bending measurements. The length of the plastic plate was much longer than the tapered length, and all of the behaviors of the interferometers were measured to minimize the influence of the differences in L between samples. The optical resolution (RES) of the OSA was set to 0.5 nm. In order to study the spectral responses of the power coupling between the input and output cores, the spectral curves from input core 1 to output core 1, namely 1–1, was called the through state, and the spectral curves from the input core 1 to the output cores 2/3/4, namely 1–2/3/4, was defined as the cross-coupling state. For the corner-core excitation scheme used, the laser signal was launched into a corner core of the FCF, which was defined as excitation core 1. Before tapering, cores 2–4 were dark, whereas only the core 1 was illuminated. No optical cross-coupling occurred, even when the FCF was bent, since the distance between the cores was far enough. When the FCF was tapered, evanescent coupling occurred among tri-cores 1–3 to excite the new modes; the new excited modes, together with the original guiding mode, are commonly supposed to form the asymmetric supermodes, as in the picture shown in Figure 1c. When FCF was further tapered, the diagonal core 4 entered the evanescent coupling region to produce new asymmetric supermodes based on a quadruple-core structure, similar to the picture shown in Figure 1d. For the supermode generation transited from a tri-core to a quadruple-core structure, a few more higher-order supermodes were excited due to the complex core structure in a smaller D of the TFCF [45]. As shown in Figure 3a, in the TFCF-1, the output spectral responses (1–2 and 1–3) were similar when respectively measured from cores 2 and 3 using a fan-out coupler. This was because core 2 and core 3 were physically identical in position and in dimension from the viewpoint of core 1. However, the output spectral responses of cores 2 and 3 were quite different from that of core 1, which we ascribed to the supermode interference. The ripples in the spectral oscillations reflected that several supermodes were excited and the high-order supermodes were advantageous to the high bending sensitivity. However, the output spectral characteristics became highly dependent on the wavelength, since the resonant dips of curves 1–2 and 1–3 were in phase at around 1370 nm and out of phase at around 1525 nm, respectively, with respect to the resonant dip and peak of curve 1–1. On the other hand, the output spectral response of core 4 at the diagonal position had an opposite phase when the wavelength was longer than 1450 nm. For the wavelengths shorter than 1450 nm, the oscillations were not clear, since the evanescent coupling occurred starting from the longer wavelengths during tapering. As shown in Figure 3a–c, the free spectral range was not found to be strongly wavelength dependent when D was substantially decreased. This phenomenon was different from the typical optical characteristics of the conventional tapered fiber couplers [46,47].

To proceed to the bending measurements, two TFCFs with a D of 13.5 μm (L = 2.18 mm) and 7.5 μm (L = 2.21 mm) were selected due to their higher sensitivity. The samples were respectively glued and mounted on a 12 mm long plastic strip. The two ends of the plastic strip were bilaterally clamped by the stainless rods fixed onto the moving blocks driving by the stepping motors. During measurement, the displacement resolution of the steeping motors was precisely controlled to be 5 μm in each incremental step for bending measurement purposes, whereas the best displacement resolution of the system was 1 μm. The spectral responses and measured sensitivity of TFCF-2 are shown in Figure 4a,b over the bending range of 0.357–1.556 m^−1^. The wavelengths at the four transmission dips (A1, B1, C1, and D1) shown in Figure 4a were measured, and their sensitivities were calculated as 9.986 nm/m^−1^, 8.66 nm/m^−1^, 10.87 nm/m^−1^, and 10.38 nm/m^−1^, respectively. In addition, the R^2^ of the fitted curve reached 0.952, 0.984, 0.97, and 0.97, respectively. The results of bending measurement of TFCF-3 are shown in Figure 4c,d. Over the bending range of 0.399–1.129 m^−1^, the sensitivity of the four dips (A2, B2, C2, and D2) were 16.12 nm/m^−1^, 10.43 nm/m^−1^, 12.04 nm/m^−1^, and 11.75 nm/m^−1^, respectively, as shown in Figure 4d. The coefficient of determination (R^2^) of the fitted curve reached 0.961, 0.976, 0.981, 0.988, respectively. As shown in Figure 4b,d, the R^2^ were on average above 0.97, which explicitly reflected a good linearity of the bending sensing versus the resonant wavelength shift. In addition, the slope of the wavelength shift was higher when D = 7.5 μm than when D = 13.5 μm, since a smaller D could lead to a wider field distribution of the supermodes. Moreover, in Figure 1d, the high-order supermodes can be implicitly observed from the interference fringes. For sensors, a higher-order mode is known to produce a higher sensitivity due to the stronger mode interaction with the external materials or the stronger OPD due to sensor deformation. Consequently, the spectral responses for the cross-coupling state are found to be more sensitive than the direct-through condition. Based on the above discussions, it was explicitly revealed that the tapered diameter of the TFCF, as well as the arrangements of the cores, were decisive parameters for the bending sensing. However, due to the commercially available FCF, the tapered diameter was experimentally investigated and found to be a main decisive parameter for bending sensing in this work. In short, the merits of using TFCF for bending sensing with enhanced sensitivity include: (1) asymmetric supermode generation based on the corner-core excitation scheme; (2) generation of the asymmetric mode field distribution for higher sensitivity; and (3) higher-order supermodes excitation in the cross-coupling states. Thus, the bending sensors based on asymmetric supermodes using TFCF had a better sensitivity than many other methods using FBGs, LPGs, offset dislocation, or interferometers [48,49,50,51].

## 5. Conclusions

In conclusion, it was experimentally demonstrated that asymmetric supermode interference based on a TFCF in a corner-core excitation scheme resulted in a high-sensitivity bending sensor. For a sensor with a tapered diameter of 7.5 μm, the best sensitivity was as high as 16.12 nm/m^−1^ within a bending range of 0.399–1.129 m^−1^, and the best extinction ratio was 16.85 dB. The asymmetric supermodes could be generated from a tri-core structure and then transit to a quadruple structure with a decreasing D. This phenomenon split the spectral characteristics into in-phase and out-of-phase at wavelengths shorter and longer than 1450 nm, respectively. The asymmetric supermodes were different from those of symmetrical supermodes in the conventional tapered MCF to produce a higher bending sensitivity. The results are promising for the development of new fiber interferometric sensors with a high sensitivity.

## Figures and Tables

**Figure 1 sensors-22-05698-f001:**
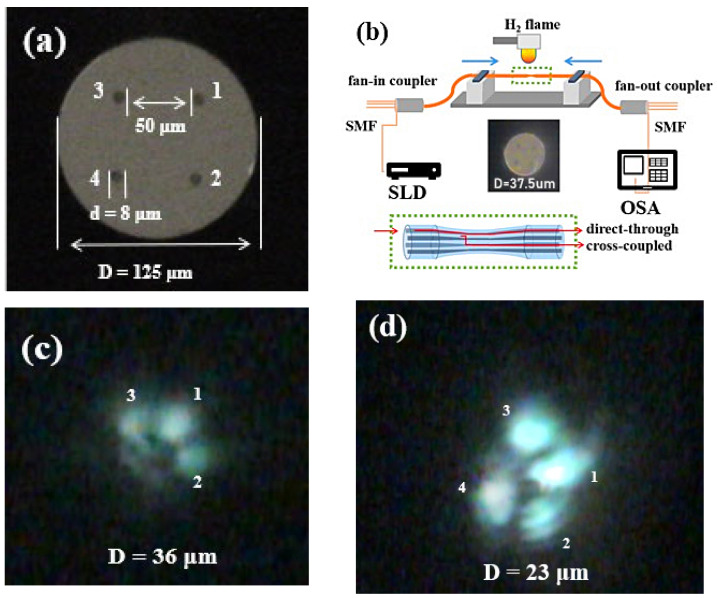
(**a**) Designation of the four cores in the TFCF interferometer, in which the number 1 represents the excitation core, and the numbers 2, 3, and 4 respectively denote cores 2, 3, and 4; (**b**) Experimental setup for fabricating TFCF bending sensors based on the corner excitation scheme. The inset picture is a cross-sectional view of the microphotographs of the TFCF when D = 37.5 μm under a 1000× CCD microscope; (**c**,**d**) Mode field distribution of the excited supermodes when (**c**) D = 36 μm and (**d**) D = 23 μm.

**Figure 2 sensors-22-05698-f002:**
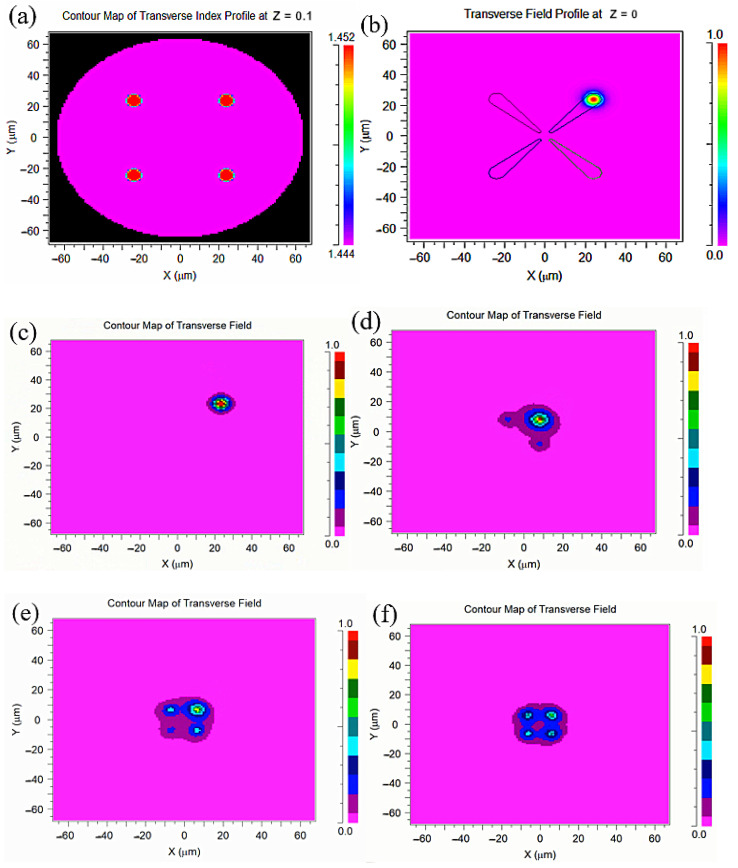
(**a**) The cross section of the FCF; (**b**) The input excitation was defined as the input in the upper right corner; (**c**–**g**) The normalized electrical field intensity profiles of supermodes in TFCF when the FCF diameter gradually decreased; (**h**) The normalized electrical field intensity profiles when the input light passed through one conical region of the TFCF to the other side.

**Figure 3 sensors-22-05698-f003:**
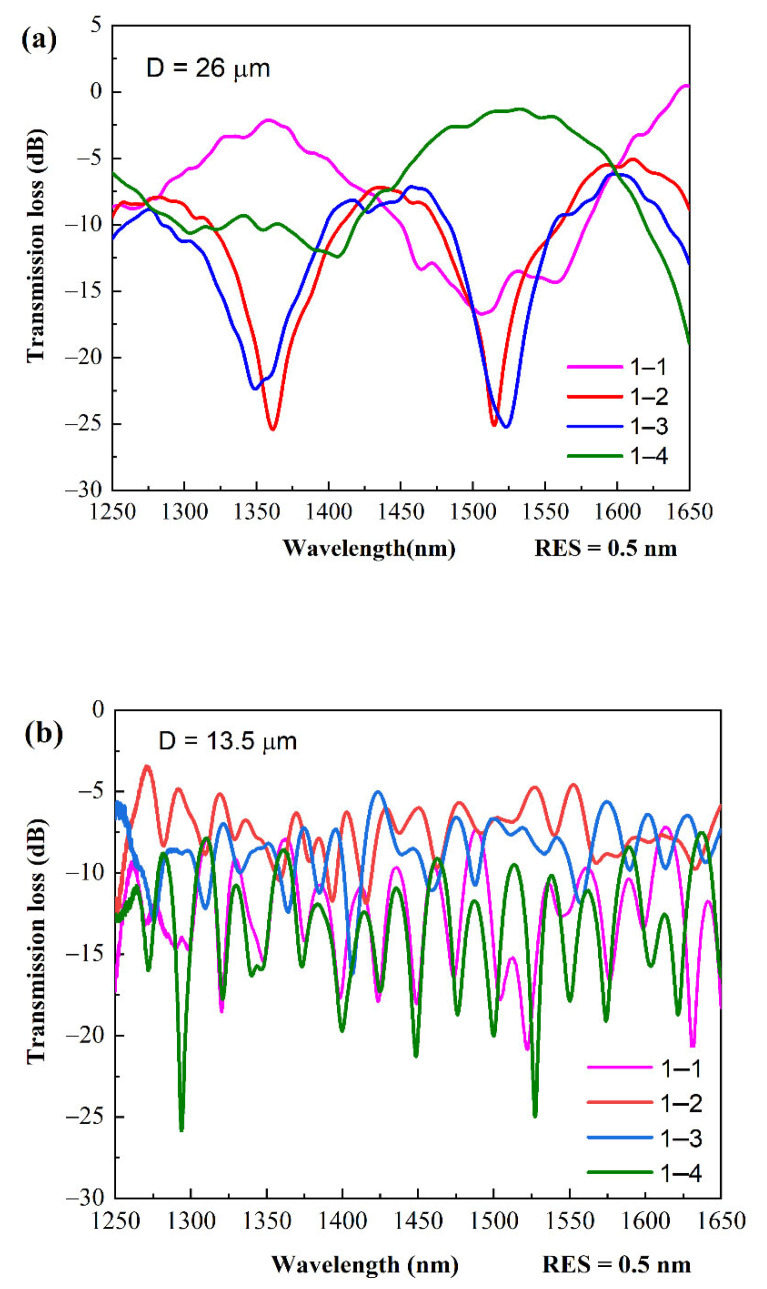
Spectral responses of the oscillating curves for: (**a**) D = 26 μm, L = 1 mm; (**b**) D = 13.5 μm, L = 2.18 mm; (**c**) D = 7.5 μm, L = 2.21 mm in the 1–1/2/3/4 states.

**Figure 4 sensors-22-05698-f004:**
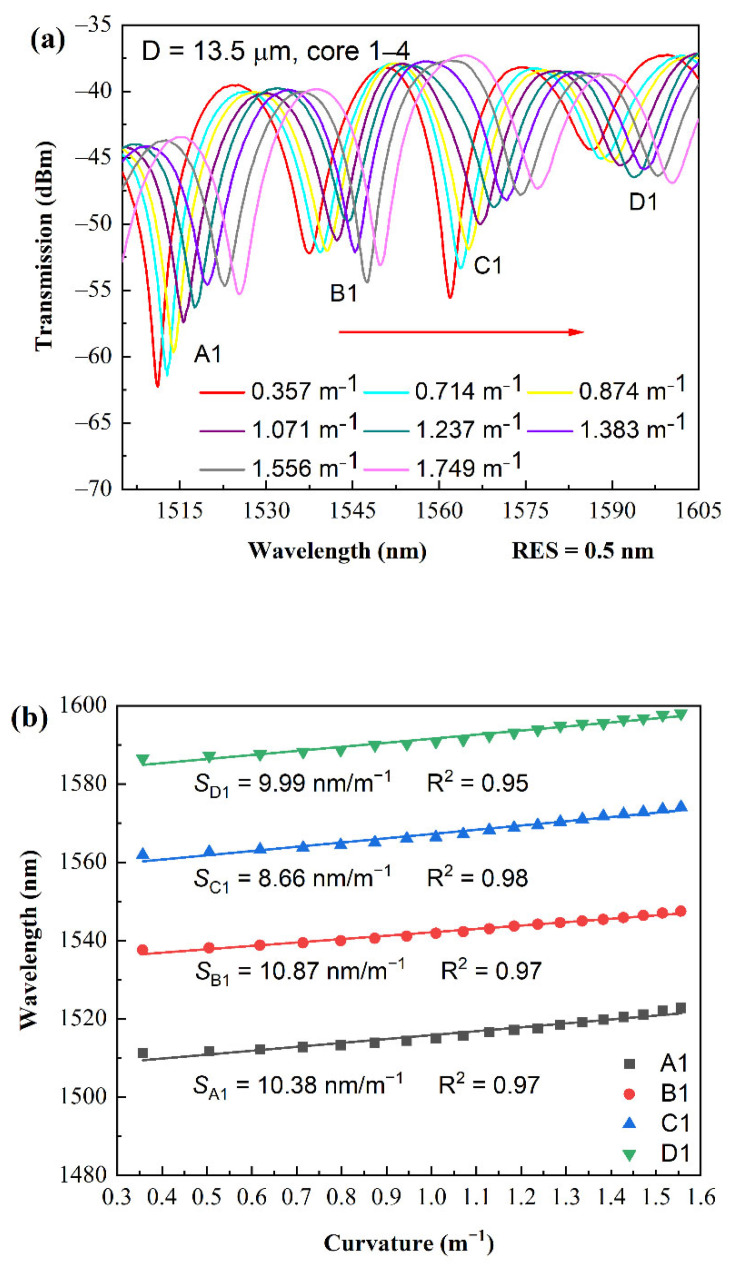
Spectral responses of the tapered FCF interferometer under different tensile bending in (**a**) TFCF-2 and (**c**) TFCF-3. Linear fitting curves of the dip wavelength shifts and the coefficients of determination (R^2^) of linear fitting in (**b**) TFCF-2 and (**d**) TFCF-3.

## Data Availability

The data that support the findings of this study are available upon request from the corresponding author. The data are not publicly available due to privacy or ethical restrictions.

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
