# Peer review of "Multicore Fiber Bending Sensors with High Sensitivity Based on Asymmetric Excitation Scheme"

_sensors, 2022, doi:10.3390/s22155698_

Round 1

Reviewer 1 Report

In this manuscript, the authors proposed a bending senor based on a tapered four-core optical fiber. The four-core optical fiber was tapered, and supermodes can be generated. Experimental results show that the maximum sensitivity of the bending sensor can reach 16.12 nm/m-1 when the tapered diameter is 7 μm. After reviewing this work, I can not recommend acceptance of this manuscript. My comments are as follows.

1.     Generally, due to the fragility of tapered fiber structures, they are rarely selected as bend sensors. At the same time, how to accurately determine the non-bending state of the structure?

2.     In fact the highest sensitivity of the bend sensor is achieved with a taper diameter of 7.5 μm instead of the 7 μm claimed by the authors in the Abstract.

3.     In Fig. 1, the position of the core 1 in Fig. 1(a) should be consistent with Fig. 1(c), Fig. 1(d) and Fig. 4.

4.     Is the heat source used by the authors in the tapering process oxyhydrogen flame? Or hydrogen flame?

5.   For mode field distribution, I recommend that the authors give a comparison of simulation and measurement results.

6.     The authors claimed that “When D is above 30 μm, the optical signal passes directly, that is, input from core 1 and output from core 1.” (line 135, page 4), however, there are theoretical or experimental results to support it.

7.     Why did the authors choose to use tapering diameters of 13.5 and 7.5 μm as experimental measurements?

8.     Why the measurement range for TFCF-2 and TFCF-3 are different?

9.     Why the authors just presented the results of TFCF-2 with core 1-1 and TFCF-3 with core 1-4?

10.   Please check the fonts and size of the figures carefully.

Author Response

In this manuscript, the authors proposed a bending senor based on a tapered four-core optical fiber. The four-core optical fiber was tapered, and supermodes can be generated. Experimental results show that the maximum sensitivity of the bending sensor can reach 16.12 nm/m-1 when the tapered diameter is 7 μm. After reviewing this work, I can not recommend acceptance of this manuscript. My comments are as follows.

  1. Generally, due to the fragility of tapered fiber structures, they are rarely selected as bend sensors. At the same time, how to accurately determine the non-bending state of the structure?

Response 1: There have been many reports about the tapered fiber structures [1-4]. And at present, there are many commercial tapered fiber products, such as couplers used in communication fields, so its fragility depends on the preparation process. After our research, the tapered fiber has strong toughness after the effective internal stress is eliminated. Our experimental results show that the diameter of the fiber can be tapered down to 1 um and still maintain good toughness and not easy to break. In addition, the lifetime of the tapered fiber can be up to 1-2 years.

There are two ways to know for sure how to remain the non-bending state.  First, a CCD with high magnification is used to observe whether the tapered fiber remains straight. In lines 242-246, We illustrate our measurement process. “The two ends of the plastic strip were bilaterally clamped by the stainless rods fixed onto the moving blocks driving by the stepping motors.” This also ensures that our tapered fiber is straight.

[1] Zheng, Yu et. al. “Highly sensitive bending sensor based on a tapered hollow core microstructured optical fiber” 2020 CONFERENCE ON LASERS AND ELECTRO-OPTICS (CLEO)

[2] Biyao Yang, et.al. “High sensitivity balloon-like refractometric sensor based on singlemode-tapered multimode-singlemode fiber” 2018, Sensors and Actuators A: physical, 281,42-47

[3] Biyao Yang, et.al. “High Sensitivity Curvature Sensor with Intensity Demodulation Based on Single-Mode-Tapered Multimode–Single-Mode Fiber” 2018, IEEE Sensors Journal, 18(3), 1094-1099.

[4] Shijie Tan, et.al.” A Large Measurement Range Bending Sensor Based on Microfiber Probe” 2019, IEEE Photonics Technology Letters, 31(24), 1964-1967.

  1. In fact the highest sensitivity of the bend sensor is achieved with a taper diameter of 7.5 μm instead of the 7 μm claimed by the authors in the Abstract.

Response 2: We have revised the description of the cone diameter in the abstract: ”When the diameter of TFCF is 7.5 μm and a tapered length is 2.21 mm, the sensitivity of the bending sensor can reach 16.12 nm/m-1”。

  1. In Fig. 1, the position of the core 1 in Fig. 1(a) should be consistent with Fig. 1(c), Fig. 1(d) and Fig. 4.

Response 3: We modified the position of core 1 in Fig. 1(a) as suggested.

  1. Is the heat source used by the authors in the tapering process oxyhydrogen flame? Or hydrogen flame?

Response 4: The heat source used in the cone process is a hydrogen flame.

  1. For mode field distribution, I recommend that the authors give a comparison of simulation and measurement results.

Response 5: The simulation results correspond to the measurement results shown in Fig. 1(c) and (d).

  1. The authors claimed that “When D is above 30 μm, the optical signal passes directly, that is, input from core 1 and output from core 1.” (line 135, page 4), however, there are theoretical or experimental results to support it.

Response 6: We observed the output spectrum in the experiment and found that when the cone diameter reached 26 μm, the transmission spectrum interfered (as shown in fig.3 (a)).  And through simulation analysis, interference begins to occur when the diameter is less than about 30 μm.

  1. Why did the authors choose to use tapering diameters of 13.5 and 7.5 μm as experimental measurements?

Response 7: This is because when we make the tapered fiber sensor, the diameter of the tapered fiber sensor produced is different due to the difference in the size of the flame and the drawing speed and time of the tapering machine.

  1. Why the measurement range for TFCF-2 and TFCF-3 are different?

Response 8: We actually tested all of the spectral data, but in this manuscript, for ease of reading, we only selected some of the spectral information.

  1. Why the authors just presented the results of TFCF-2 with core 1-1 and TFCF-3 with core 1-4?

Response 9: Thank the reviewer for reminding us.  We are very sorry that there is a mistake in our drawing.  In fact, there are the spectral responses of two samples with core 1-4. We had revised the figure 4 (a) and (c) in the revised manuscript.

  1. Please check the fonts and size of the figures carefully.

Response 10: We have carefully checked the graphs in the manuscript.

Reviewer 2 Report

The authors experimentally demonstrated the use of the asymmetric supermodes in the tapered four-core optical fiber (TFCF) for  bending sensors. The mode excitation was exhibited and analyzed. This work is interesting for the readers from multi-core fibers and fiber sensors. The manuscript could be accepted if the following matters are properly addressed.

1 In the tapering process, what is the best tapered diameter for the sensing performance? 

2 How about the stability of the sensor? Considering the inter-core cross-talk and mode interference.

3 The author should discuss the temperature influence on the sensor in the experiments.

4 The English language needs to be improved, maybe by the use of professional proofreading service. 

Author Response

The authors experimentally demonstrated the use of the asymmetric supermodes in the tapered four-core optical fiber (TFCF) for  bending sensors. The mode excitation was exhibited and analyzed. This work is interesting for the readers from multi-core fibers and fiber sensors. The manuscript could be accepted if the following matters are properly addressed.

1 In the tapering process, what is the best tapered diameter for the sensing performance? 

Response 1: The sensing performance of the tapered fiber optic sensor we made increases gradually with the decrease of the diameter of the taper, so the best performance diameter obtained by our experiments is 7.5 μm.

2 How about the stability of the sensor? Considering the inter-core cross-talk and mode interference.

Response 2: The supermode interference proposed in our manuscript is the superposition of inter-core cross-talk. So we need more inter-core cross-talk to improve the sensitivity of our sensors. The effect of mode interference depends on the optical path difference, which is determined by the size and distance of the inter-core. Therefore, for the same tapered fiber sensor, it has high stability.

3 The author should discuss the temperature influence on the sensor in the experiments.

Response 3: The whole process of the experiment is carried out in a clean room with constant temperature and humidity, so the cross-correlation of temperature is avoided.

4 The English language needs to be improved, maybe by the use of professional proofreading service.

Response 4: We have consulted native English experts for professional proofreading service as suggested.

Reviewer 3 Report

A diagram of the bending apparatus would be helpful to the reader. 

Page 1: Abstract: The diameter of 7 microns is stated. Is this a measurement or an approximation. please clarify in the text. 

Page 4: please define lambda

Page 4: please define OSA and all other acronyms at first use. 

Page 4: line 168, please correct the word "supermodel" to the intended word of "supermode"

page 6 line 209: delete word between "coupling" and "occurred"

Figure 3: fonts too small to read comfortably. Please add labels between the port coupling name and the waveforms. 

Figure 4: Fonts too small to read comfortably. 

Author Response

A diagram of the bending apparatus would be helpful to the reader. 

The experimental set-up for fabricating TFCF bending sensors based on the corner excitation scheme as shown in fig.1 (b).

Page 1: Abstract: The diameter of 7 microns is stated. Is this a measurement or an approximation. please clarify in the text. 

Response 1: We have revised the description of the cone diameter in the abstract: ”When the diameter of TFCF is 7.5 μm and a tapered length is 2.21 mm, the sensitivity of the bending sensor can reach 16.12 nm/m-1”.

Page 4: please define lambda

Response 2: Thank you for reviewer’s reminder. We had added definition in the revised manuscript.

Page 4: please define OSA and all other acronyms at first use. 

Response 3: We define OSA on line 120 of the page 3. ”The spectral responses of the 4 cores were individually connected to the optical spectrum analyzer (OSA) through the fan-out coupler to record the spectral characteristics.”

Page 4: line 168, please correct the word "supermodel" to the intended word of "supermode"

Response 4: We had corrected the word "supermodel" to the intended word of "supermode" in line 169 and 172.

page 6 line 209: delete word between "coupling" and "occurred"

Response 5: We had removed the word between "coupling" and "occurred".

Figure 3: fonts too small to read comfortably. Please add labels between the port coupling name and the waveforms. 

Response 6: We had revised it.

Figure 4: Fonts too small to read comfortably

Response 7: We had revised it.

Round 2

Reviewer 1 Report

My comments have been properly addressed, and I recommend acceptance of this manuscript.

Author Response

Thank you for your valuable comments.